# Cancer Screening Program Delivered by Community Health Workers for Chinese Married Immigrant Women in Korea

**DOI:** 10.3390/ijerph19116655

**Published:** 2022-05-30

**Authors:** Jiyun Kim, Yuna Paik, Seungmi Park

**Affiliations:** 1School of Nursing, Gachon University, Incheon 21936, Korea; jkim@gachon.ac.kr (J.K.); ksyuna1@gachon.ac.kr (Y.P.); 2Department of Nursing Science, Chungbuk National University, Cheongju 28644, Korea

**Keywords:** community health workers, immigrant woman, cancer screening, health education

## Abstract

This quasi-experimental study used a cancer prevention program delivered by community health workers (CHWs) as an intervention to improve health literacy and increase screening uptake. The intervention group was educated by trained CHWs and received information about the nearby hospitals. After education, participants received telephone counseling once a month for 6 months. In the intervention group, CHWs met the participants individually and delivered the CD-ROM containing conversation scenarios with voice during the cancer screening test. The control group was provided educational materials related to breast and cervical cancers and a booklet containing information on mammography and Pap test. This study assessed the difference in mammography and Pap tests between the intervention and control groups. The results showed that the participants’ knowledge improved, and the rate of cancer screening tests did not decrease in the intervention group. Therefore, it is necessary to develop and apply various programs that train CHWs and apply them to immigrant women to promote health-related behaviors under the health system that they are not familiar with while living in another country.

## 1. Introduction

Multicultural marriage is a historically long-standing phenomenon occurring in Korea since the late 1980s [1]. This phenomenon was initiated to address the gender imbalance between men and women living in rural areas [2]. The government encouraged finding a bride abroad so that rural bachelors could get married [2,3], as Korean women frequently prefer to live in city centers where educational and professional opportunities are increased [2]. This immigration movement brings from abroad young women of childbearing age seeking to build a family in Korea [4]. As a result, in 2020 the multicultural marriage rate was estimated at 7% [5].

Married immigrant women are exposed to health risk factors and experience numerous health issues. Some of these issues are related to pregnancy (e.g., infertility and spontaneous abortion), childbirth [6], as well as other conditions such as anemia, allergic disease, gastric and duodenal ulcer, asthma, and uterine myomas [7]. The prevalence of gynecological diseases among married immigrant women that required active management was found to be 11.6% [8]. Nevertheless, married immigrant women have low medical consultation rates [9]. Factors found to impact the low utilization of health services by immigrant women in Korea include the financial burden of medical fees [8], difficulty using public transportation [8], lack of social support [10], and the lack of language fluency [11].

Young women in their 20s are becoming middle-aged women. Therefore, it is necessary to develop and implement programs to promote screening for cancer prevention and to increase screening uptake by immigrant women. In Korea, national cancer screening is conducted for stomach cancer, colorectal cancer, breast cancer, liver cancer, lung cancer, and cervical cancer according to age [12]. Women over the age of 20 can get a Pap smear every two years, and those over 40 can get a free mammography every two years [12]. Furthermore, in a study that analyzed factors affecting mammography and Pap test in married immigrant women living in Korea, the number of screening behaviors increased if they had relevant educational experience [9].

Community health workers (CHWs) are trained paraprofessionals who deliver programs to community members to ensure appropriate health education and care in their communities [13,14,15]. There are many studies that employed CHWs when implementing programs for a population with linguistic and cultural disparities, such as female married immigrants [16,17,18]. The delivery of the intervention by CHWs is particularly effective and cost-effective because CHWs and participants share their language and understand the culture of cancer screening [19,20,21]. In addition, promoting health literacy in practicing healthy behaviors, such as cancer screening tests, can promote changes in cognition and attitude [22].

The decision to involve Chinese women in the role of CHWs was twofold: (1) cultural sensitivity, as a high proportion of immigrant women are from Chinese backgrounds [5]; (2) a high proportion of immigrant Chinese women are in the target age group of 30–40 years [23].

This study used a cancer prevention program as an intervention to increase health literacy and screening uptake by Chinese female married immigrants. This study assessed the difference in mammography and Pap test knowledge and uptake between the intervention and control groups. Through this study, it will be possible to plan and implement a more appropriate program for female married immigrants in Korea.

## 2. Materials and Methods

### 2.1. Research Design and Ethical Considerations

This quantitative quasi-experimental randomized study used a cancer prevention program delivered by trained CHWs as an intervention in the community. This study assessed the difference in mammography and pap tests between the intervention and control groups. The institutional review board of Gachon University reviewed and approved this study (1044396-201504-HR-028-01).

### 2.2. Sample Size and Power Calculation

The number of samples for McNemar’s test was obtained by referring to the results of a previous study [24]. The odds ratio and the proportion expected to change due to the one-tailed, 1.95 effect size at an alpha of 0.05, 80% power, and 47.8% of the population changing because of the intervention, 126 participants would be needed to detect the effect. When the dropout rate was calculated as 30%, the required number of participants was 163.

### 2.3. Participants and Study Setting

#### 2.3.1. Study Procedure

This study consisted of a procedure whereby the CHW who received training recruited participants and provided a cancer prevention program and telephone counseling. The period from the CHW recruitment to the 6-month follow-up of the participants was from April 2015 to May 2016.

#### 2.3.2. CHWs Recruitment and Study Training

A senior researcher visited multicultural centers to recruit CHWs directly or used notice boards from multicultural centers during the period of April 2015 to May 2016. A CHW is a woman who is fluent in Korean and received training from the research team when she was assigned to a group (intervention or control group). The CHWs received six hours of training over 2 days to conduct the study procedures. The study team developed training materials based on cancer-related educational materials presented on the National Cancer Center website [12] and the National Health Insurance Corporation website [25]. The training was provided by a bilingual research team and covered topics such as the main contents of education, including the introduction of the concept and role of CHWs, contents on breast and cervical cancer, prevention tests and procedures for breast and cervical cancer, data collection procedures for the participants, education on the prevention of cancer for women—centered on the ability to understand medical information and practice for telephone counseling. Seventeen CHWs were trained in June and October 2015. Sixteen CHWs were active during the study period, as one woman decided not to perform CHW before the participants’ recruitment. Once a month, the CHW and research team met to check the study’s progress and provide feedback on their activities. The CHWs in the control group were also trained for two hours to confirm participant eligibility and collect questionnaires.

#### 2.3.3. Recruiting Participants and Delivering Intervention by CHWs

Participants were recruited from four multicultural centers in the metropolitan area. Two multicultural centers were selected for the intervention group and the other two were selected for the control group. The multicultural centers provide various support services such as Korean language education, translation services, life coaching services, and improved multi-cultural awareness [26]. Therefore, married immigrant women visit multicultural centers to get the benefits and gather to meet the women who immigrated from the same country and engage in emotional exchanges together [4]. Participants were eligible to be involved in the study if they were (1) Chinese women living in Korea after marrying Korean men, (2) aged 30–64 years, (3) did not undergo a Pap test or mammography within the last year, (4) could read and write Korean, and (5) agreed to participate in the research. In this study, 183 participants were included in the final analysis, after excluding those who dropped out of the group. The baseline data for both groups (intervention and control) was collected during the period from June 2015 to December 2015. The follow-up data for both groups (intervention and control) was collected during the period from January to June 2016. The process of recruiting and dropping out of the study is shown in Figure 1.

The intervention group was educated by trained CHWs and received information about the nearby hospitals. Educational materials for study participants were developed to improve their ability to understand medical information by referring to the educational materials produced in the previous study [22]. In the intervention group, CHW met the participants individually and delivered the compact disk read-only memory (CD-ROM)-containing conversation scenarios with voice during the cancer screening test. The educational materials were a picture book containing both Korean dialogue and Chinese translation and a CD-ROM provided with dialogue in Korean. The program contents included several scenarios illustrating the steps to undertake cancer screening tests, such as reception desk, conversation with doctors, and physical examination. Two nursing professors and two obstetrics and gynecology nurses reviewed the intervention scenarios. After the education by CHWs on the participants was completed, once a month for six months, the CHWs provided telephone counseling to share their thoughts and help them decide on the examination. The control group was offered educational materials related to breast and cervical cancers and a booklet containing mammography and Pap test information.

### 2.4. Outcome Measures

Self-reported questionnaires measured mammography knowledge and Pap test knowledge. To confirm suitability and correctness, translation and back-translation techniques were used [27]. Therefore, these questionnaires were translated into Chinese, and two bilingual nursing doctors translated and reviewed them. First, two bilingual CMIW were tested while they responded to the questionnaire. Then, the two PhDs modified the questionnaire several times so that the participants in this study could easily read and respond.

The mammography knowledge questionnaire consisted of 18 questions and verified its validity and reliability [28]. The questionnaire examples were “It is not necessary to look at your breasts during breast self-examination” and “Squeezing the nipple is necessary for a good examination”. Cronbach’s α in this study was 0.909. The Pap test knowledge questionnaire consisted of 10 items [29], with a Cronbach’s α of 0.867. The cervical cancer knowledge questionnaire examples were “Women that have never been pregnant will not get cervical cancer” and “If one smokes heavily, the risk for cervical cancer increases”.

At the 6-month follow-up, the CHWs collected the information on uptake of mammography and Pap tests via participants’ self-report questionnaires.

### 2.5. Data Analysis

The characteristics of the study subjects were described using t-test and χ^2^ tests. Repeated-measures ANOVA was used to confirm the difference in mammogram and Pap test knowledge between the intervention group and the control group. The McNemar test was used to analyze the difference in screening tests between the intervention group and the control group. All analyses were performed using SPSS 26.

## 3. Results

The participants’ average age was 38.33 ± 7.03 years, the intervention group was 38.28 ± 6.39 years, and the control group was 38.40 ± 7.73 years. Education levels of the intervention group show that24 people (13.1%) graduated from middle school or less, 47 (25.7%) graduated from high school, and 26 (14.2%) graduated from college or higher. In the control group, 19 people (10.4%) graduated from middle school or less, 37 (20.2%) graduated from high school, and 30 graduated from college or higher. The average length of residence in Korea was 8.03 ± 3.94 years, 7.77 ± 4.30 years in the intervention group, and 8.31 ± 3.49 years in the control group. As for occupation, 133 people (72.7%) had jobs, 75 in the intervention group (41.0%), and 58 (31.7%) in the control group. Regarding total income, 85 (46.4%) had less than 200 million won, 57 (31.1%) had 200–300 million won or less, and 41 (22.4%) had more than 300 million won. Ten (5.5%) participants had no children, 55 (30.1%) had one child, and 32 (17.5%) had two or more children in the intervention group.

There was a statistically significant difference in the mammography knowledge scores between two groups, with an average of 8.29 ± 4.09 points in the intervention group and 6.44 ± 4.22 in the control group (t = −3.002 *p* = *0*.003). Although the Pap test knowledge score was 4.77 ± 2.60 in the intervention group and 4.20 ± 2.60 in the control group, there was no statistically significant difference between the two groups (t = −1.493 *p* = 0.137).

Regarding mammography uptake before the program, 14 (23.0%) participants in the intervention group and 10 (16.4%) in the control group said they had done it. Regarding the Pap test uptake before the program, 20 (10.9%) participants in the intervention group and 20 (10.9%) in the control group said they completed the program. The similarity of the intervention and control groups regarding the general characteristics was confirmed by the homogeneity test (*p* > 0.005) (Table 1).

As for the program’s effect on mammography knowledge, there was a significant difference between the measurement period and group interaction (F = 9.279, *p* = 0.003), between the groups (F = 9.297, *p* = 0.003), and time points (F. = 9.297, *p* = 0.003). Therefore, this program has a significant effect on mammography knowledge.

The program’s effect on Pap test knowledge was not significantly different between the measurement period and group interaction (F = 0.079, *p* = 0.779); however, there was a statistically significant difference between the groups (F = 4.073, *p* = 0.045) and time points (F = 15.698, *p* ≤ 0.001) (Table 2).

The number of participants who underwent a mammography before and after program implementation increased from 14 (23.0%) to 19 (31.1%) in the intervention group, but the difference was not statistically significant (*p* = 0.267). Conversely, the number of participants who underwent mammography was 10 (16.4%), and the number decreased to 0 (0%) in the control group, with a statistically significant difference (*p* = 0.002).

The number of participants who underwent Pap test before and after program implementation increased from 20 (10.9%) to 23 (23.7%) in the intervention group, but the difference was not statistically significant (*p* = 0.690). However, in the control group, the number of participants who received the Pap test decreased from 20 (10.9%) to 9 (10.5%), with a statistically significant difference (*p* = 0.019) (Table 3).

## 4. Discussion

In high-income countries, CHWs have mainly linked vulnerable groups, including immigrants, to chronic disease health services [30]. For example, while many studies on intervention programs that promote female cancer screening, such as Pap test and mammography of immigrant women, mainly focused on Hispanic and African Americans, few studies have been conducted on migrant women in Asia [31].

CHWs play an important role to connect the healthcare system and community members [32]. A study that investigated the satisfaction of CHW among immigrants from Bangladesh, the Philippines, and India showed a high-reliability rate (83.5%) [31]. In addition, a high percentage of respondents reported that they could discuss health problems that they could not directly discuss with a doctor [31]. It seems that CHWs are a workforce capable of providing services tailored to the unique needs of immigrants with respect to cultural congruence and trust from immigrants.

In a previous study that conducted a program to promote CHW-led health literacy for Korean women who immigrated to the United States, the study results showed that their breast cancer knowledge and mammography and Pap test screening rates increased [22]. While in the control group mammography and Pap tests showed a significantly lower test rate, confirming the need for CHW intervention to promote health behaviors such as cancer screening. In this study, the intervention that CHWs navigated through screening and provided counseling helped Chinese married immigrant women to undergo cancer screening. However, it is necessary to understand the characteristics of Chinese women through the result that the control group in this study had significantly lower mammography and Pap test acceptance rates. For example, Chinese people are less likely to go to the hospital if they are not sick [33], so they may feel less need for preventive screening tests. Chinese women feel vulnerable from an unfamiliar health care system [34]. Chinese women showed higher health-seeking behaviors as their language proficiency improved in the country they emigrated to [35,36]. To improve access to screening services for Chinese female immigrants, it is necessary to educate CHWs on culturally sensitive content, including raising awareness of screening tests and removing language barriers.

In a study conducted on a program using CHW-led multimedia to increase the screening rate of cervical cancer for immigrants in Asia [37], health beliefs about the usefulness of screening increased, and perceived barriers decreased. An educational strategy using multimedia, which is effective in populations with low literacy, can be considered [33]. It is hoped that future research will include an intervention program using CHW-led multimedia for immigrant women. In addition, it is necessary to strengthen training and provide opportunities for feedback during activities so that CHWs operating in Korea can fulfill their roles when delivering mediation to female married immigrants [33].

The limitations of this study are as follows. Since the intervention using CHW was conducted only for Chinese women residing in Korea, its application to women migrants from other countries is limited. Since the intervention was implemented in four cities in a metropolitan area, it is difficult to generalize the results to multicultural women in rural areas of Korea. Since the intervention was provided and the follow-up period was only 6 months, it is a short period to examine the effectiveness of the intervention. In the future, it will be necessary to apply an intervention that enhances knowledge about cancer screening examination for a longer period to more participants to understand the effect. It is necessary to follow-up for a longer period in the future and conduct a study to deliver interventions to various married immigrants using CHWs. In addition, since there are cases of using CHWs in other health promotion areas such as diabetes management [38] and physical activity promotion [39], we propose a study that provides interventions for immigrants living in Korea.

## 5. Conclusions

This study provides cancer screening intervention delivered by CHWs to CMIW living in Korea. The results showed that the participants’ knowledge improved, and the rate of cancer screening tests did not decrease in the intervention group. Therefore, it is necessary to develop and apply various programs that train CHWs and apply them to immigrant women to promote health-related behaviors under the health system that they are not familiar with while living in another country.

## Figures and Tables

**Figure 1 ijerph-19-06655-f001:**
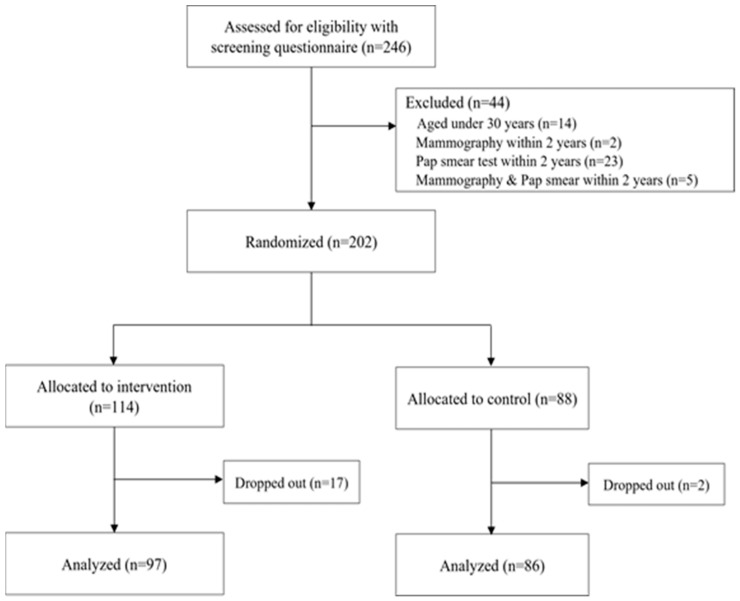
Flowchart of participant selection.

**Table 1 ijerph-19-06655-t001:** Comparison of participants’ general characteristics and outcome variables.

	Intervention	Control	Total	t/χ^2^	*p*
Age (years, range 30–64)	38.28 ± 6.39	38.40 ± 7.73	38.33 ± 7.03	−0.112	0.911
Education					
Middle school graduate or less	24 (13.1)	19 (10.4)	43 (23.5)	1.401	0.496
High school graduate	47 (25.7)	37 (20.2)	84 (45.9)		
College or more	26 (14.2)	30 (16.4)	56 (30.6)		
Length of stay in Korea (year, range 1–22)	7.77 (4.30)	8.31 ± 3.49	8.03 ± 3.94	−0.926	0.356
Employment					
Employed	75 (41.0)	58 (31.7)	133 (72.7)	2.240	0.134
Unemployed	22 (12.0)	28 (15.3)	50 (27.3)		
Income level (Korean Won)					
200 million or less	37 (20.2)	48 (26.2)	85 (46.4)	5.858	0.053
300 million or less	34 (18.6)	23 (12.6)	57 (31.1)		
Over 300 million	26 (14.2)	15 (8.2)	41 (22.4)		
Number of children					
0	10 (5.5)	7 (3.8)	17 (9.3)	0.361	0.835
1	55 (30.1)	48 (26.2)	103 (56.3)		
2 or more	32 (17.5)	31 (16.9)	63 (34.4)		
Mammography knowledge	8.29 ± 4.09	6.44 ± 4.22	7.42 ± 4.24	−3.002	0.003
Pap test knowledge	4.77 ± 2.60	4.20 ± 2.60	4.50 ± 2.61	−1.493	0.137
History of screening					
Underwent mammography	14 (23.0)	10 (16.4)	24 (39.3)	0.108	0.742
Underwent a Pap test	20 (10.9)	20 (10.9)	40 (21.9)	0.186	0.667

**Table 2 ijerph-19-06655-t002:** Comparison of mammography knowledge and Pap test knowledge (n = 183).

Variable	Groups	Pre	Post	Sources	F (*p*)		Post-Pre t (*p*)
Mammography knowledge	Intervention (97)	8.29 ± 4.09	9.41 ± 3.28	Group	9.297	0.003	4.643 (<0.001)
	Control (86)	6.44 ± 4.22	8.24 ± 4.16	Time	9.297	0.003	3.780 (<0.001)
				Group × Time	9.297	0.003	
Pap test knowledge	Intervention (97)	4.77 ± 2.60	5.58 ± 2.17	Group	4.073	0.045	3.996 (<0.001)
	Control (86)	4.20 ± 2.60	4.90 ± 2.46	Time	15.698	<0.001	2.361 (0.020)
				Group × Time	0.079	0.779	

**Table 3 ijerph-19-06655-t003:** Comparison of undergoing mammography and Pap test.

		Pre	Post	*p*
Underwent mammography (%)	Intervention	14 (23.0)	19 (31.1)	0.267
	Control	10 (16.4)	0 (0)	0.002
Underwent Pap test (%)	Intervention	20 (10.9)	23 (23.7)	0.690
	Control	20 (10.9)	9 (10.5)	0.019

## Data Availability

Data are available on request from the corresponding author.

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
