# Peer review of "Cancer Screening Program Delivered by Community Health Workers for Chinese Married Immigrant Women in Korea"

_ijerph, 2022, doi:10.3390/ijerph19116655_

Round 1

Reviewer 1 Report

The article entitled: “Cancer screening program delivered by Community Health workers for Chinese Married Immigrant Women in Korea” shows the importance of educational interventions in public health.

The article could be suitable to be published in IJERPP, and should be interesting for the readers. After carefully reviewed the work I consider the article has potential to be accepted, since it is an important example of the importance of educational programs to improve public health’s aspects. I believe the authors highlighted the main limitations of the study and provide strong evidence of the importance of interventions. Moreover, I consider that the study is a good contribution to provide help and education to minorities, which should be excluded from public health programs.  

Author Response

The article entitled: “Cancer screening program delivered by Community Health workers for Chinese Married Immigrant Women in Korea” shows the importance of educational interventions in public health.

The article could be suitable to be published in IJERPH and should be interesting for the readers. After carefully reviewing the work, I consider the article has the potential to be accepted since it is an important example of the importance of educational programs to improve public health. I believe the authors highlighted the main limitations of the study and provide strong evidence of the importance of interventions. Moreover, I consider that the study is a good contribution to providing help and education to minorities, which should be excluded from public health programs.

>>Thank you for your opinion. In the discussion, we added the contents about intervention for immigrants.

Reviewer 2 Report

Dear Authors, I have listed in the letter attached my concerns about your manuscript. Thank you

Author Response

Reviewer: #2

Effective screening programs lead to early disease detection and improved health outcomes; however, their impact will depend significantly on the participation rates of the target population. Among minorities, the screening uptake is known to be low. Therefore, implementing strategies to improve the uptake of screening is critical to improving the health outcomes of these populations. This manuscript evaluated a cancer screening program delivered by community health workers to immigrant women in Korea. While the manuscript is relevant to cancer care, it requires considerable changes, especially for an international audience to appreciate this work. The manuscript is very succinct, in the introduction and discussion. Below are listed the points requiring revision:

  1. Abstract: I suggest re-structuring the abstract and revising the aim.

I would suggest starting the text from the sentence on line 10: ’This quasi-experimental study used a cancer prevention program as an intervention to improve health literacy and increase screening uptake. The intervention group was educated by trained CHWs and received information about the nearby hospitals. After education, participants received telephone counselling once a month for 6 months. In the intervention group, CHW met the participants individually and delivered the CD-ROM containing conversation scenario with voice during the cancer screening test. The control group was provided educational materials related to breast and cervical cancers and a booklet containing information on mammography and Pap test. This study assessed the difference in mammography and Pap tests between the intervention and control groups. The results showed that the participants’ knowledge improved, and the rate of cancer screening tests did not decrease in the intervention group. Therefore, it is necessary to develop and apply various programs that train CHWs and apply them to immigrant women to promote health-related behaviors under the health system that they are not familiar with while living in another country.’

>>Done. Thank you very much for your suggestion.

  1. Introduction

The very first issue with the Introduction section is that authors appear to assume that readers know about the historical and local circumstances surrounding the immigration of women in bearing age to Korea. I suggest that the authors expand on the information and clarify the immigration phenomenon. Also, I would recommend reviewing the language used, some statements appear to portray Chinese female immigrants’ in a less favourable light.

2.1. Page 1, on line 26, my suggestion is that the authors first describe the multicultural marriage phenomenon. So in first paragraph, perhaps start with something in these lines: ‘In Korea multicultural marriage is a historically long standing phenomenon being described since ??? and started because of ???. This immigration movement brings to Korea young women in child bearing age seeking to build a family. In 2020, the multicultural marriage rate was estimate as 7%.’

>>Done from lines 26 to 32. 

2.2. Page 1, lines 26-32, move this paragraph down to become the second paragraph, ‘Married immigrant women……

>> Done.

2.3. Page 1, line 26-27, what are the health problems that these women are exposed: increased number of pregnancies? Is pregnancy a health problem?

>> Done from lines 34 to 35.

2.4. Page 1, on line 28, ‘a survey of these diseases’: what diseases are the authors referring to? Pregnancy or childbirth are not a disease?! Please clarify.

>> Done from lines 35 to 40. We cited other references to clarify the health problems experienced by immigrant women.

2.5. Page 1, on line 28, ‘female diseases’: what diseases? Please specify.

>> Done from lines 37 to 39.

2.6. Page 1, on line 29, replace ‘examinations’ by ‘consultations’.

>> Done from lines 41 to 42.

2.7. Page 1, on lines 30-31, there is a missing ‘in’ between the words ‘impact in the low…’; Add the word ‘services’ after health and remove ‘rate’ after health: ‘….health services…’

>>Done

Remove ‘of’ after health services and add ‘by’: ‘…health services by immigrant women….’; Add the word ‘financial’ before burden: ‘…the financial burden of medical fees….’;  Clarify ‘difficulty using transportation’ immigrants don’t know where to catch a bus, don’t drive?

>>Done from lines 41 to 43.

Clarify ‘human networks’, do the authors mean social support of family, community, services?? 

>>Done.

2.8. Page 1, on line 40, there should be a link between this line and the introduction of the CHWs as the facilitators of the education program.

>>Done from lines 55 to 59.

2.9. Page 2, lines 58-61. I suggest re-structuring the sentence: ‘This study used a cancer prevention program as an intervention to increase health literacy and screening uptake by Chinese female married immigrants. This study assessed the difference in knowledge and uptake of mammography and Pap smear tests between the intervention and control groups’.

>> Done lines from 72 to 75.

  1. Materials and Methods:

3.1. Page 2, lines 65-66, research design and ethical considerations. Please consider reviewing, it appears to be a confusion about the aim of the study and the strategies used to achieve the aim. My understanding is that the aim is to increase health literacy and screening uptake. The strategies used were health education and counselling. My suggestion: ‘This quasi-experimental randomized study used a cancer prevention program delivered by CHWs as an intervention.

>>Done lines from 80 to 82. We revised as you suggested. Thank you for your consideration.

3.2. ‘Participants’, Page 2, on line 70, this section needs re-structuring as participants ‘Chinese women living in Korea’ and ‘community health workers (CHWs)’ were all presented in the same section. I suggest having a section for participants ‘Chinese women’ only and add information about the settings: how many multicultural centres and what services are provided in these centres:

‘Participants and study settings’

Participants were recruited by CHWs from multicultural centres (how many centers??) in Seoul (?), Korea. These centres provide XXX services to support immigrants (men and women?). Participants were eligible to be involved in the study if they: (1) were Chinese women living in Korea after marrying Korean men, (2) aged 30–64 years, (3) did not undergo Pap test or mammography within the last 1 year, (4) could read and write Korean and (5) agreed to participate in the research.

>>Done the lines from 93 to 113. We moved and changed the contents as you suggested.

3.3. Page 2, line 73, insert new sub-heading ‘Community Health Workers (CHWs): Recruitment and study training’. Start the paragraph from line 73 and then move the paragraph from page 3, lines 114-120 to page 2

‘Senior researchers visited multicultural centers to recruit CHWs directly or used notice boards(?) from multicultural centers during the period of April 2015 to May 2016. A CHW is a woman who is fluent in Korean and received training from the research team when she was assigned to an intervention group.

The CHWs received 6 h of training over 2 days to provide research interventions. The training was provided by a bilingual research team and covered topics such as the main contents of education, including the introduction of the concept and role of CHWs, contents on breast and cervical cancer, prevention tests and procedures for breast and cervical cancer, data collection procedures for the participants, education on the prevention of cancer for women-centered on the ability to understand medical information and practice for telephone counselling. Seventeen CHWs were trained in June and October 2015. CHWs were active during the study period, as one woman decided not to perform CHW before the participants’ recruitment. Once a month, the CHW and research team met to check the study’s progress and provide feedback on their activities. The CHWs in the control group were also trained for 2 h to confirm participant eligibility and collect questionnaires.

>>Done the lines 92 to 113.

3.3. Page 3, on line 96, ‘Intervention development’ I suggest changing the sub-heading to ‘Intervention resources and implementation’ as it explains how the intervention and control groups were assigned and the information provided to each group. It does not explain how the research team developed the education resources but what the resources were. Include the sentence in page 3, lines 112 and 113, in this section: ‘For the intervention group, 17 CHWs were trained to provide research interventions, including health literacy training in breast and cervical cancer screening and monthly telephone counselling.’

>>Done. We input this content in the 2.3.2. Recruiting participants and delivering intervention by CHWs

3.4. Page 3, on line 97, the first sentence is unclear, were the intervention groups assigned in two cities and the control groups in four cities? Please clarify.

>>Done the lines 117 to 118

3.5. Page 4, on line 126, I suggest adding the word ‘outcome’ and making the sub-heading plural ‘Outcome Measures’

>>Done.

3.6. Page 4, lines 127-128: This section needs more re-structuring, there is a lot of information but very challenging to follow, consider reviewing the paragraph.

For example first sentence, I suggest: ‘The outcome variables, knowledge and health literacy, were assessed by two tools: knowledge of cancer screening questionnaire and health literacy assessment tool.’

>>Done. We used two measurements such as mammography knowledge and pap test knowledge, not health literacy. Therefore, we deleted health literacy in this section.

3.7. Page 4, line 128-129, the authors mention: ‘These methods’, my understanding is that the authors refer to the tools or questionnaires, is confusing to follow the description of the outcome measures

>> We revised the contents and explained the two knowledge questionnaires.

3.8. Page 4, line 135-136, in the sentence: ‘The knowledge of the cancer screening questionnaire is for two separate questionnaires…..’, do the authors mean ‘The knowledge of cancer screening was assessed through two separate questionnaires?’.

>> Yes. We used two separate questionnaires. One is mammogram and the second is Pap test. We revised the contents.

3.9. Page 4, line 145, add a subheading ‘Sample size and power calculation’, under this subheading start the paragraph by moving from Page 2, lines 79-84, to this section. My suggestion, please check if the information is correct:

‘The results of a previous study [20] were used to The number of samples estimate the sample size in each group (intervention and control group) and allow testing changes using McNemar’s test. The odds ratio and the proportion expected to change due to the one-tailed, 1.95 effect size at an alpha of .05, 80% power, and 47.8% of the population changing because of the intervention, 126 participants would be needed to detect the effect. When the dropout rate was calculated as 30%, the required number of participants was 163.’

>> Done. We moved this content to 2.2 for guiding the study process.

3.10. Page 4, on line 145, remove the ‘t’ after the word ‘Data’ in the subheading.

>>Done. Thank you.

3.11. Page 4, lines 146-150, consider revising. ‘The characteristics of the study subjects were described using t-test and χ2 tests. Repeated-measures ANOVA was used to confirm the difference in mammogram…..

>>Done.

3.12. Page 4, line 149, there is a misspelling in the name ‘McNamar’.

>>Done. Thank you.

  1. Discussion: overall the discussion is mainly focused on the role of CHWs and not the results for the participants. Many programs worldwide promote screening, utilize resources to facilitate uptake and often offer free screening and despite this, uptake often remains suboptimal, mirroring health service uptake in general with pockets of the population showing poor coverage. The discussion need to consider what other factors could have contributed for the low participation rate in the screening programs. There should be the recognition of the importance of cultural, social and economic factors on health and health behaviours. Arguably, these requirements are more important in screening where there is a lack of transport to seek health care (as mentioned in the introduction that transport is an issue for the study population) as the population targeted is asymptomatic. Ultimately, the authors may consider reviewing the training delivered to CHWs; providing to CHWs a training that is culturally sensitive and point to actions that improve the access to screening services for Chinese female immigrant married to Korean males. Increasing knowledge is not always sufficient to motivate changes in behaviour, especially for adults.

>>Done.

Reviewer 3 Report

This is an interesting study whose results can be translated in public health interventions. The authors should provide more context about reproductive healthcare challenges among the studied population (in case relevant data are available), provide the full name of the University granting the ethical clearance in the methods section and revise the manuscript's language (for instance "Datat" to "Data" Analysis in one of the headings of the methods section. The authors acknowledge the limited timeframe of their study, it would be useful to outline how their protocol could be adapted to a long - term health promotion intervention and study.

Author Response

Reviewer: #3

This is an interesting study whose results can be translated in public health interventions.

  1. The authors should provide more context about reproductive healthcare challenges among the studied population (in case relevant data are available).

>>Done from lines 33 to 44.

  1. provide the full name of the University granting the ethical clearance in the methods section.

>>Done.

  1. revise the manuscript's language (for instance "Datat" to "Data" Analysis in one of the headings of the methods section.

>>Done.

  1. The authors acknowledge the limited timeframe of their study; it would be useful to outline how their protocol could be adapted to a long-term health promotion intervention and study.

>>Done from lines 270 to 272.

Reviewer 4 Report

1. introduction 25 - I consider it necessary to include a paragraph on health policy, particularly preventive and screening policy at the level of women's medicine, in the context where the study is carried out, so that the research methodology and results presented can be better understood.
2. 2.2 Materials and methods 63. I propose that the authors describe the method that guided the study (qualitative, quantitative or mixed?) should also clearly identify the unit or units of analysis (hospitals, clinics, other health centres).
3. 2.3. Development of the intervention 96. The authors should present the pre-defined criteria for the selection of the study respondents from both the intervention and control groups so that it is clear who was involved in the study.
4. 2.5. Measurement 126
"Two kinds of measurements, knowledge of cancer screening questionnaires and 127
health literacy assessment tools, were selected in this study for outcome variables". I propose that the authors present these instruments in a footnote, or describe them synthetically.
5. 5. Conclusions 242. The author should improve the conclusion by leaving recommendations both for new studies and at the level of health policy for women.

In a study of this nature and in the social and cultural context in which it is carried out, I consider it pertinent that the author reflects at some point on the culture of life of these women.

Author Response

Reviewer: #4

  1. introduction 25 - I consider it necessary to include a paragraph on health policy, particularly preventive and screening policy at the level of women's medicine, in the context where the study is carried out, so that the research methodology and results presented can be better understood.

>>Done from the lines 47 to 50.

  1. 2.2 Materials and methods 63. I propose that the authors describe the method that guided the study (qualitative, quantitative or mixed?) should also clearly identify the unit or units of analysis (hospitals, clinics, other health centres).

>>Done.

  1. 2.3. Development of the intervention 96. The authors should present the pre-defined criteria for the selection of the study respondents from both the intervention and control groups so that it is clear who was involved in the study.

>>Done from the line 124 to 127.

  1. 2.5. Measurement

"Two kinds of measurements, knowledge of cancer screening questionnaires and 127

health literacy assessment tools, were selected in this study for outcome variables". I propose that the authors present these instruments in a footnote, or describe them synthetically.

>>Done. The health literacy tool was not used in this paper, and we deleted that in this section.

  1. 5. Conclusions 242. The author should improve the conclusion by leaving

recommendations both for new studies and at the level of health policy for women.

>>Done from lines 272 to 275.

In a study of this nature and in the social and cultural context in which it is carried out, I consider it pertinent that the author reflects at some point on the culture of life of these women.

>>Done from lines 248 to 252.

Round 2

Reviewer 2 Report

Please find my comments attached. Best wishes

Author Response

Responses to Reviewers

Title: Cancer Screening Program Delivered by Community Health Workers for Chinese Married Immigrant Women in Korea

Thank you for sending us the comments of the reviewer. We thank the reviewer for your careful comments and suggestions to improve the quality of the paper. Our responses to each comment are provided below. Changes made in the text are shown with track change, except for changes made solely to improve the use of English.

Reviewer: #2

Introduction

Line 28. Insert ‘occurring’ before the word since

>>Done in line 25.  

Lines 29-31 I suggest re-phrasing this paragraph: My suggestion: This phenomenon was initiated to address the gender imbalance between men and women living in rural areas. The government encouraged rural bachelors to find a bride abroad so that they could get married, [2,3] as Korean women frequently prefer to live in city centers where educational and professional opportunities are increased.

>>Done from lines 26 to 32.

Line 32. Remove ‘is’ before the word immigration;

>>Done in line 32.

Line 34. Replace ‘to’ by ‘in’ before the word Korea

>>Done in line 33.

Line 34. Rephrase the sentence to: ‘As a result, in 2020 the multicultural marriage rate was estimated at 7%.’

>>Done in line 34.

Lines 36 – 44 Rephrase the sentence. My suggestion: Married immigrant women are exposed to health risk factors and experience numerous health issues. Some of these issues are related to pregnancy (e.g. infertility and spontaneous abortion), childbirth, as well as other conditions such as anemia, allergic disease, gastric and duodenal ulcer, asthma, and uterine myomas. The prevalence of gynecological diseases among married immigrant women that required active management was found to be 11.6%.

>>Done from lines 35 to 40.

Lines 52-53 - Rephrase the sentence. Therefore, it is necessary to develop and implement programs to promote screening for cancer prevention and to increase screening uptake by immigrant women.

>>Done from lines 53 to 54.

Lines 59-60 – Remove this sentence

>>Done from lines 60 to 61.

Lines 63-64 – Replace the words ‘program recipients’ by ‘Community members’. My suggestion: Community health workers (CHWs) are trained paraprofessionals who deliver programs to community members to…..

>>Done in line 63.

Line 65 – Did the authors mean ‘implementing’ instead of ‘planning’?

>>Done in line 65.

Lines 67-69 - Rephrase the sentence. I am not sure if the meaning was kept in my suggestion?? Please check if that is what the authors mean. My suggestion: The delivery of the intervention by CHWs, is particularly effective and cost-effective because CHWs and participants share their language and understand the culture towards cancer screening.

>>Done from lines 66 to 68.

Lines 72-79 - I think the point in here is to establish why CHWs with Chinese background were chosen, but the paragraph is confusing. I suggest rephrasing. My suggestion: The decision to involve Chinese women in the role of CHWs was twofold: 1) cultural sensitivity: as a high proportion of immigrant women are from Chinese backgrounds; 2) A high proportion of immigrant Chinese women are in the target age group of 30-40 years.

>>Done from lines 72 to 75.

Lines 92-93 - Insert the words ‘Knowledge’ and ‘uptake’. This study assessed the difference in mammography and Pap tests knowledge and uptake between……

>>Done from lines 86 to 87.

Materials and methods

Lines 104-107 - The authors included here what the intervention was and the heading is about CHW recruitment and training My suggestion is create another heading for ‘procedure’ and insert the text

>>Done from lines 105 to 109.

Lines 104-107 - as suggested previously and adding the text authors included in the review, the paragraph could be in these lines. My suggestion: ‘Senior researchers visited multicultural centers to recruit CHWs directly or used notice boards(?) from multicultural centers during the period of April 2015 to May 2016. A CHW is a woman who is fluent in Korean and received training from the research team when she was assigned to a group (intervention or control group). The CHWs received 6 h of training over 2 days to conduct the study procedures.

>>Done from lines 116 to 124.

Line 158 - I suggest authors consider including an initial sentence. Participants were recruited from 4 multicultural centres (N=?, out of how many?)

>>Done in line 139.

Line 159 – Remove the words ‘We Selected.

>>Done from lines 138 to 139.

Line 159 – Add ‘were selected’ Two multicultural centers were selected for the intervention group and the other two were selected for the control group.

>>Done in line 140.

Line 165 - Not sure why is this sentence here?

>>We deleted.

Line 170 – Add the word ‘after’ …final analysis, after excluding …..

>>Done in line 151.

Lines 171-173 - Rephrase the sentence

>>Done from lines 152 to 155.

Lines 184 – 185 - Rephrase the sentence

>>Done from lines 167 to 169.

Lines 215 – 216 – I am not sure if the participants completed a questionnaire or if the CHW collected what the participants reported?

>>Done from lines 195 to 197.

Results

Lines 237 – 238 - There was a statistically significant difference on the mammography knowledge scores between the two groups.

>>Done from lines 217 to 218.

Lines 241 – 243 – Please remove this sentence from this section.

>>Done.

Line 245 – Please add the word ‘uptake’

>>Done in line 225.

Lines 247 – Please add the word ‘uptake’

>>Done in line 227.

Lines 248 – 250 – The statement about the homogeneity of the two groups needs re-phrasing

>>Done from lines 228 to 230.

Discussion

Lines 275 – 276 I suggest rephrasing first sentence

>>Done from lines 255 to 257.

Line 280 – Add the word ‘important’ Instead of ‘residents at the front of the community’ use ‘community members’

>>Done from lines 261 to 262.

Line 283 – Replace ‘answered’ by ‘reported’

>>Done in line 265.

Line 285 – replace ‘they’ by ‘CHWs’ Remove the words ‘function as’

>>Done in line 267.

Line 292 – Initiate the sentence with ‘While in the control group’….,

>>Done in line 272.
